# Amyloid β_1–42_ Oligomers Induce Galectin–1^S8^ O–GlcNAcylation Leading to Microglia Migration

**DOI:** 10.3390/cells12141876

**Published:** 2023-07-17

**Authors:** Alazne Arrazola Sastre, Miriam Luque Montoro, Francisco Llavero, José L. Zugaza

**Affiliations:** 1Achucarro Basque Center for Neuroscience, Science Park of the UPV/EHU, Sede Building, 3rd Floor, Barrio de Sarriena s/n, 48940 Leioa, Spain; alazne.arrazola@ehu.eus (A.A.S.); miriamluquem@gmail.com (M.L.M.); 2Department of Genetics, Physical Anthropology and Animal Physiology, Faculty of Science and Technology, UPV/EHU, Barrio de Sarriena s/n, 48940 Leioa, Spain; 3IKERBASQUE, Basque Foundation for Science, Plaza Euskadi 5, 48009 Bilbao, Spain

**Keywords:** amyloid β_1–42_ oligomers, OGT, O-GlcNAcylation, Galectin–1, Galectin–1^Serine 8^–O–GlcNAcylation, Gal–1^S8A^, microglia, migration

## Abstract

Protein O–GlcNAcylation has been associated with neurodegenerative diseases such as Alzheimer’s disease (AD). The O–GlcNAcylation of the Amyloid Precursor Protein (APP) regulates both the trafficking and the processing of the APP through the amyloidogenic pathway, resulting in the release and aggregation of the Aβ_1–42_ peptide. Microglia clears Aβ aggregates and dead cells to maintain brain homeostasis. Here, using LC-MS/MS, we revealed that the Aβ_1–42_ oligomers modify the microglia O-GlcNAcome. We identified 55 proteins, focusing our research on Galectin-1 protein since it is a very versatile protein from a functional point of view. Combining biochemical with genetic approaches, we demonstrated that Aβ_1–42_ oligomers specifically target Galectin–1^S8^ O–GlcNAcylation via OGT. In addition to this, the Gal–1–O–GlcNAcylated form, in turn, controls human microglia migration. Given the importance of microglia migration in the progression of AD, this study reports the relationship between the Aβ_1–42_ oligomers and Serine 8–O–GlcNAcylation of Galectin–1 to drive microglial migration.

## 1. Introduction

Protein O–GlcNAcylation is a dynamic posttranslational modification (PTM) that occurs on thousands of nucleocytoplasmic and mitochondrial proteins that control important cellular processes such as transcription, translation, metabolism or signal transduction [1,2]. O–GlcNAcylation in proteins occurs at serine and threonine residues, which, in turn, could also be phosphorylated. In an unbalanced situation, one PTM could be favored at the expense of the other [3]. Thus, disruptions in the O–GlcNAcylation/phosphorylation balance have been associated with cancer [4], neurodevelopmental disorders [5], and neurodegeneration, such as Alzheimer’s Disease (AD) [6,7,8]. Regarding AD, different studies have demonstrated that in this pathology, some proteins are hyper–O–GlcNAcylated while others lose their O–GlcNAcylation [6,7,8,9].

AD is characterized by extracellular amyloid plaques, mainly constituted by the Aβ_1–42_ peptide, as well as by intracellular neurofibrillary tangles formed by the hyperphosphorylated Tau protein [10]. The accumulation and aggregation of these molecules in extra and intracellular spaces result in dystrophic neurites and the loss of neurons and synapses, which causes a loss of memory, spatiotemporal disorientation and behavioral changes, among other symptoms [10]. Recent studies have associated O–GlcNAcylation with this disease; in fact, when the amyloid precursor protein (APP) is O–GlcNAcylated at threonine 576, this PTM promotes the trafficking and processing of the APP through the amyloidogenic pathway, which results in the generation of the Aβ_1–42_ peptide [9]. However, O–GlcNAcylation in the Tau protein could have a neuroprotective role [6]. Its O–GlcNAcylation at serine and threonine residues prevents Tau hyperphosphorylation via its competition with the phosphosites [11,12]. In addition, O–GlcNAcylation could also play a neuroprotective role by inhibiting necrosis in AD. Therefore, the toxicity or the protective role of O–GlcNAcylation in AD could be protein-specific [13].

The presence of reactive glia in AD around extracellular amyloid plaques has been well established, with microglia examples of such cells [14]. Microglia are responsible for an inflammatory response that provokes neuronal death, and it is also recruited to amyloid plaques in order to phagocytose Aβ_1–42_ oligomers [15,16,17]. For this, microglia first require a whole mechanism of migration towards the plaques. The process mentioned is especially relevant in the initial stages of the disease as it prevents the formation of amyloid plaques [16,17]. However, with aging, microglia lose their phagocytic function, and the clearance of Aβ_1–42_ is insufficient, favoring the pathology of AD [18]. In fact, microglia from old AD-mice express low levels of Aβ-binding receptors, as well as low levels of Aβ-degrading enzymes compared to microglia from young AD-mice [19]. Hence, owing to its functional duality, with a pro-inflammatory function on the one hand and its beneficial phagocytic function on the other, the role of microglia in AD is still controversial.

Here, we have revealed that Aβ_1–42_ oligomers modify the protein O-GlcNAcylation pattern in human microglia, identifying 55 proteins. One of these is galectin-1 which is specifically O–GlcNAcylated on serine 8. This O–GlcNAcylation, in turn, controls microglia migration mediated by Aβ_1–42_ oligomers. Given the importance of migration in the progression of AD, in this study, we describe a possible mechanism to control this cellular response through Gal–1 Ser^8^ O–GlcNAcylation.

## 2. Materials and Methods

### 2.1. Materials

Wheat germ agglutinin (WGA)-agarose beads, Thiamet G, OSMI-I and collagen type I, 2XYT media, isopropyl-β-D-thiogalactopyranoside (IPTG), Tris, EDTA, 1,1,1,3,3,3-hexafluoro-2-propanol (HFP), phenylmethanesulfony (PMSF), IGEPAL^®^ CA-630, sodium orthovanadate (Na_3_VO_4_), sodium fluoride (NaF), Dimethyl sulfoxide (DMSO), paraformaldehyde (PFA), Protein A-Sepharose^TM^CL-4B beads were from Merck KGAA, Headquarters of the Merck Group, Frankfurter Straße 250. Darmstadt, 64293, Germany. Nickel-agarose beads, Dulbecco’s Modified Eagle Medium (DMEM) high glucose, Ham’s F12 medium, fetal bovine serum (FBS), penicillin and streptomycin were from Thermo Fisher Scientific, 168 Third Avenue, Waltham, MA USA.

### 2.2. Production and Purification of Amyloid-β_1–42_ (Aβ_1–42_) and Oligomers Preparation

The production and purification of Aβ_1–42_ oligomers were based on [20] with some modifications. Bacterial cultures were grown overnight in 10 mL 2XYT media with 100 μg/mL of ampicillin at 37 °C and at 220 rpm. After approximately 16 h, the bacterial cultures were diluted to 1:10 and incubated at 30 °C for 3 h at 235 rpm. After the incubation, protein expression was induced with 1 mM IPTG at 13 °C for 24 h at 235 rpm. Finally, the bacterial suspension was centrifuged at 4500 rpm for 10 min, and the supernatant was discarded. The pellet was resuspended in 10 mL of buffer A (10 mM Tris-HCl pH 8, 1 mM EDTA). The bacterial suspension was sonicated for 2 min on ice in a UP100H Ultrasonic Processor (Hielscher, Oderstraße 53, 14513 Teltow, Germany) with ½ horn and 50% duty cycle. This solution was then centrifuged at 18,000× *g* for 10 min at 4 °C, and the supernatant was discarded. The pellet containing the inclusion bodies was resuspended in a 10 mL buffer. It was subjected to other 2 cycles of sonication, centrifugation and resuspension in buffer A. After 3 whole cycles, the last pellet was resuspended in 5 mL of 8 M urea in buffer A, and the solution was incubated on ice for 15 min. After that period, the solution containing urea-solubilized inclusion bodies was sonicated and centrifuged, and finally, the supernatant containing the Aβ_1–42_ peptide was collected and diluted with 15 mL buffer A.

The Aβ_1–42_ peptide was then purified by ion exchange chromatography in a NGC Quest 10 Liquid Chromatography System (Life Science, 1000 Alfred Nobel Drive, Hercules, CA 94547, USA). The Bio-Scale^TM^ Mini UNOsphere^TM^ Q Cartridge (Life Science) column was first equilibrated with buffer A. The 20 mL sample containing Aβ_1–42_ peptide was then loaded onto the column. The column was washed with buffer A, and the protein was eluted by ion exchange with a gradient of 0 to 300 mM NaCl in buffer A. The eluted fractions containing the Aβ_1–42_ peptide was collected and subjected to another purification and concentration protocol. These fractions were passed through a 30 kDa molecular mass cut-off filter Amicon Ultra-15 membrane PLTK Ultracel-PL 30 kDa (Merck KGAA) in order to remove proteins that were bigger than 30 kDa. For this purpose, the filter was first washed with H_2_O at 4000× *g* for 20 min. Then, the protein solution was added, and it was centrifuged at 4000× *g* for 20 min. The fraction that passed the filter (≤30 kDa) was collected. The ≤30 kDa solution was passed through a 3 kDa molecular mass cut-off filter Amicon Ultra-15 membrane PLBC Ultracel-PL 3 kDa (Merck KGAA) to discard the proteins that were smaller than 3 kDa. For this second purification step, the filter was first washed with H_2_O at 4000× *g* for 30 min, and then, the ≤30 kDa solution was added and centrifuged at 4000× *g* for 60 min. We discarded the solution that passed the filter (≤3 kDa). In order to remove the salt coming from the chromatography of the solution containing Aβ_1–42_ monomers, we added miliQ H_2_O to the fraction, which did not pass the filter, and we centrifuged it at 4000× *g*. After 60 min, the fraction that had not passed the filter was collected, which was concentrated in approximately 400 µL of H_2_O.

The filtered volume was then lyophilized In a Virtis Lyophilizer (Millrock Technology, Inc., 39 Kieffer Lane, Kingston, NY 12401, USA) overnight. After approximately 18 h, the lyophilized Aβ_1–42_ peptide was weighted. After that, stock peptide films of the Aβ_1–42_ peptide were prepared based on the protocol reported by Klein [21]. Briefly, the lyophilized Aβ_1–42_ peptide was solubilized in cold HFP at a final peptide concentration of 1 mM. The solubilization process was carried out in an Actor Ultrasonic Bath (Lovango, Pintor Roig i Soler, 14, 08916 Badalona, Spain) for approximately 2 h until the solution turned clear and colorless. Finally, the solution was aliquoted, and the HFP was left to evaporate in the fume hood for 1 h. Once evaporated, the tubes were transferred to a SpeedVac SC110 Vacuum Concentrator (Thermo Fisher Scientific, 168 Third Avenue, Waltham, MA, USA) and dried down for 10 min, obtaining a thin clear peptide film at the bottom of the tubes. The aliquots were kept at −80 °C until use. The day before the experiments, one aliquot of the peptide film was dissolved in 5 mM of DMSO in Ham’s F12 medium in a volume such that the final peptide concentration was 100 µM. The solution was left for 18 h at 4 °C in order to form the globular oligomers, also known as amyloid β-Derived Diffusible Ligands (ADDL).

### 2.3. Plasmid Construct and TAT-OBD-Gal-1 Protein Purification

Complementary DNA encoding for a 20-amino acid peptide (residues 1–20) of Gal–1 that contained serine 8 (called OGT-binding domain or OBD-Gal-1) was amplified by a polymerase chain reaction using the following oligonucleotides: Fw: CTCGAGATGGCTTGTGGTC Rv: GGAATTCCCTCGCCTCGC cloned in vector p*TAT-HA*. The p*TAT-OBD-Gal-1* plasmid was transformed into *Escherichia coli* (BL21) and proteins expressed by IPTG induction. The recombinant TAT–OBD–Gal–1 protein was purified by sequential chromatography on nickel-agarose beads and anion exchange chromatography [22].

### 2.4. Site-Directed Mutagenesis

The mutagenesis of Gal-1^S8A^ was performed with the QuickChange II Site-Directed Mutagenesis Kit according to the manufacturer’s instructions (Agilent Technologies, 5301 Stevens Creek Blvd., Santa Clara, CA 95051, USA). Original plasmid *mDsRed-Gal-1* was a gift from Michael Davidson (Addgene plasmid #55831; http://n2t.net/addgene/55831; RRID: Addgene_55831, Addgene Headquarters, 490 Arsenal Way, Suite 100, Watertown, MA 02472, USA). It was amplified with the following oligonucleotides *mDsRed–Gal–1^S8A^* Fw (5’-GCTTGTGGTCTGGTCGCCGCCAACCTGAATCTCAAACC-3′) and *mDsRed–Gal–1^S8A^* Rv (5′-GGTTTGAGATTCAGGTTGGCGGCGACCAGACCACAAGC-3′).

### 2.5. Cell Culture

Immortalized human microglia (Cat no T0251, Applied Biological Materials Inc. #1-3671 Viking Way, Richmond, BC V6V 2J5, Canada) were cultured in DMEM high-glucose, supplemented with 10% FBS and 100 U/mL of penicillin and 100 μg/mL of streptomycin. Cells were seeded in plates that had previously been coated with 0.01% (*v*/*v*) collagen type I and incubated at 37 °C in an atmosphere with 5% CO_2_. For the experiments, cells were washed 3 times with PBS before being deprived of 1% FBS for 24 h. On the day of the experiments, the cells were untreated or treated with 5 μM Aβ_1–42_ oligomers as described [23]. To examine the effects of OSMI–I and Thiamet G, these inhibitors were added at 25 μM and 10 μM, respectively, for 16 h before their stimulation with Aβ_1–42_ oligomers.

### 2.6. Microglia Transient Transfection

Microglia were washed twice with PBS and resuspended in DMEM high glucose without FBS. The 4 × 10^6^ cells/200 μL medium were then transferred to an electroporation cuvette of 0.4 cm (Merck KGAA) containing between 5 and 10 μg cDNA of interest as indicated in the Section 3. The electroporation was carried out at 260 V and 950 μF in a Gene Pulser Xcell^TM^ electroporator (Life Science) [24]. After that, electroporated cells were collected, and DMEM with FBS was added. The transfected cells were incubated for 48 h in the incubator at 37 °C before the experiments [24]. Microglia were also transfected with Lipofectamine^TM^ 3000 (Thermo Fisher Scientific) according to the manufacturer’s instructions 48 h prior to the experiments.

### 2.7. Wheat Germ Agglutinin-Affinity Precipitation and Protein Identification by LC-MS/MS

The cells were either stimulated or not with 5 µM Aβ_1–42_ oligomers at 37 °C for 30 min [23,25] and washed twice with cold PBS. Next, the cells were lysed with a RIPA buffer (50 mM Tris-HCl pH 7.4, 1% IGEPAL^®^ CA-630, 0.25% sodium deoxycholate, 150 mM NaCl, 1 mM EDTA, 1 mM PMSF, 1 mM Na_3_VO_4_, 1 mM NaF and protease inhibitor cocktail ((P8340) from Merck KGAA) for 20 min at 4 °C in an orbital shaker. Then, the lysates were centrifuged at 13500 rpm for 10 min at 4 °C, and the supernatants (500 µg protein) were incubated with previously washed WGA-agarose beads for 1 h at 4 °C in an orbital shaker. After that, the complexes were washed three times with a binding buffer at 4000 rpm for 1 min at 4 °C. The O-GlcNAcylated proteins were dissociated from the WGA-agarose beads by the addition of SDS sample loading buffer (Merck KGAA, Reference 70607) and samples were analyzed by SDS-PAGE.

For the O–GlcNAcylated proteomic analysis, 6.4 × 10^7^ cells and WGA-agarose beads were used. After purification, the O–GlcNAcylated proteins were eluted from the complexes by the addition of 20 mM GlcNAc in a binding buffer for 10 min at 4 °C in an orbital shaker. The samples were then centrifuged at 4000 rpm for 1 min at 4 °C, and the supernatants (3 mg protein per condition) were collected and analyzed by LC-MS/MS in the Proteomics General Service (SGIker) of the University of the Basque Country (UPV/EHU). The Q Exactive mass spectrometer (Thermo Scientific) coupled to the Easy-nLC 1000 nanoUPLC System (Thermo Scientific) chromatograph was used. The resulting spectra were processed with a Proteome Discoverer 1.4 (Thermo Scientific). Swiss-Prot human database, which was used for peptide identification.

WGA purification was also used for the specific analysis of Gal–1 O–GlcNAcylation. Microglia were transfected with 10 µg of either a plasmid encoding for *mDsRed–Gal–1* or *mDsRe–-Gal–1^S8A^* by electroporation. Cells were serum-deprived for 24 h and were either stimulated or not with 5 µM Aβ_1–42_ oligomers for 30 min. Lysates (500 µg protein) were obtained in the RIPA buffer containing the OGA inhibitor Thiamet G, and O–GlcNAcylated proteins were purified with WGA-agarose beads at 4 °C. The affinity complexes were then washed three times with RIPA + Thiamet G, and the proteins were eluted. The eluted proteins and the total input fractions were analyzed by SDS-PAGE, and the presence of Gal–1 was determined by a Western blot; proteins were transferred to PVDF membranes (Life Science) and then were blocked with 5% milk in TBST before finally being blotted using a specific anti-DsRed antibody (Takara Bio Europe SAS, 34 rue de la Croix de fer, 78100 Saint-Germain en Laye, France. Reference 632496, dilution 1:1000. Antibodies were incubated for 1 h at RT).

### 2.8. In Vitro O-GlcNAcTransferase (OGT) Assay

The in vitro OGT assay from microglia was based on [26]. Briefly, microglia were transfected with *pDest*-*N-Myc-OGT* by electroporation. After 24 h, transfected cells were serum-deprived for an additional 24 h. On the day of the experiment, the cells were either stimulated or not with 5 µM Aβ_1–42_ oligomers for 30 min, and they were lysed with a RIPA buffer containing the O–GlcNAcase (OGA) inhibitor (10 µM Thiamet G). In total, 500 µg of protein per condition were incubated for 2 h at 4 °C with anti-Myc antibody bound to protein A-Sepharose beads. Immunocomplexes were washed, and the transferase reaction was made by adding 100 μL of 1 mM UDP–GlcNAc and 500 ng GST–Gal–1 as the substrates. The reaction was incubated at 37 °C for 3 h and was stopped by the addition of SDS sample loading buffer (Merck KGAA, Reference 70607). The proteins were separated by SDS-PAGE and OGT activity and measured by analyzing GST–Gal–1 O–GlcNAcylation by SDS-PAGE and Western blot. The proteins were transferred to PVDF membranes (Life Science) before being blocked with 5% milk in TBST and finally blotted using specific antibodies: anti-O–GlcNAc (Merck KGAA, reference 05-1245, dilution 1:1000, anti-GST (Merck KGAA, reference G7781, dilution 1:50,000) and anti-Myc (Merck KGAA, reference M4439, dilution 1:5000. Antibodies were incubated for 1 h at RT).

### 2.9. Scratch Wound Healing Assay

Scratch wound healing was performed based on [27]. The cells were seeded 24 h prior to the experiment and were serum-deprived with 1% FBS and the experiment was conducted in this serum-free medium to ensure that there was no proliferation during the migration assay. A scratch was performed on the cell monolayer, and cells were either stimulated or not with 5 µM Aβ_1–42_ oligomers. In experiments with inhibitors, these compounds were added 16 h prior to the stimulation, as indicated in the Section 3. Photographs of the wound were taken in four random fields with the 4× objective in an EVOS™ Digital Color Fluorescence Microscope (Life Science) at the beginning of the experiment and after 24 h of incubation at 37 °C. The percentage of migration was calculated in the Fiji-ImageJ by measuring the width of the wound before (0 h) and after (24 h) the cells had migrated.

### 2.10. Transwell Migration Assay

A transwell assay was conducted based on [27] using the Costar Transwell System with an 8-µm pore size polycarbonate membrane (Merck KGAA). In total, 250,000 human microglia cells were serum-deprived with 1% FBS for 24 h, which were seeded in the upper chamber for the 200 µL serum-free medium and in the presence or the absence of 5 µM Aβ_1–42_ oligomers. In total, 600 µL of the serum-free medium was placed in the lower chamber. The cells were incubated for 24 h at 37 °C. The microglia that had not migrated were removed from the upper side of the membrane with a cotton swab, and cells that had migrated to the lower side were fixed with 4% PFA for 10 min. Migrated cells were then stained with Mayer’s Hematoxylin for 20 min. Four randomly chosen areas were photographed with the 10× objective in an EVOS™ Digital Color Fluorescence Microscope (Life Science), and these cells were counted in Fiji-ImageJ (National Institutes of General Medical Sciences, 45 Center Drive MSC 6200, Bethesda, MD 20892-6200, USA) in order to calculate the percentage of migrating cells.

### 2.11. Statistical Analysis

All data were expressed as the mean ± S.E.M. Statistical analyses were performed using GraphPad Prism statistical software (versión 5.0; GraphPad Software, 225 Franklin Street. FI. 26, Boston, MA 02110, USA). Comparisons between the experimental groups were made using Mann–Whitney statistical analysis. To determine the significance between data means, (* *p* < 0.05, ** *p* < 0.01, *** *p* < 0.001).

## 3. Results

### 3.1. Aβ_1–42_ Oligomers Increase Protein O-GlcNAcylation in Human Microglia

Firstly, we investigated if the protein O-GlcNAcylation pattern was modified in the presence of Aβ_1–42_ oligomers in human microglia. For this, 1 × 10^7^ cells were treated or not with 5 μM Aβ_1–42_ oligomers for 30 min before being washed and lysed. Then, O–GlcNAcylated proteins were purified with wheat germ agglutinin (WGA) agarose beads and analyzed by SDS-PAGE followed by Western blotting. WGA-purified proteins from Aβ_1–42_ oligomers-stimulated microglia showed an increased O–GlcNAcylation pattern (Figure 1A, lane 2 compared to lane 1). The total lysates (input) were used as an internal control (Figure 1A, lanes 3, 4). The quantification of three independent experiments demonstrated that the stimulation resulted in a 1 ± 0-fold increase in the microglial protein O–GlcNAcylation compared to the untreated control (Figure 1A, bar graph).

These results prompted us to identify the specific proteins that were differentially O–GlcNAcylated in the Aβ_1–42_ oligomers-stimulated microglia. With this purpose, the cells were either stimulated or not with 5 µM Aβ_1–42_ oligomers for 30 min, and O–GlcNAcylated proteins were purified and analyzed by LC-MS/MS as described in the Section 2. As shown in the Venn diagram of Figure 1B, the results from LC-MS/MS identified 44 proteins in untreated cells, where 55 proteins were identified in the Aβ_1–42_ oligomers-stimulated cells with 182 proteins in both conditions (Figure 1B and Appendix A).

Of these 55 proteins, Galectin–1 (Gal–1) was chosen among several others owing to its functional versatility. In fact, Gal–1 controls many signaling cascades both in an intra- and extracellular manner, with potential implications in the control of microglial migration [28,29]. Analyzing the specific peptides of Gal–1 obtained by LC-MS/MS, one of them was the ACGLVA**S**NLNLKPGECLR peptide. This peptide was located at the N-terminal from aa 2 to aa 19 and contained a serine, Ser^8^, which could be O–GlcNAcylated. This result suggested that the O–GlcNAcylation in this specific Ser could play an important role in the function of the protein.

Next, we examined whether Aβ_1–42_ oligomers induced Gal–1 O–GlcNAcylation in human microglia and secondly verified if this PTM occurred on Ser^8^. Therefore, microglia were transfected with 10 µg of either a plasmid encoding for *mDsRed–Gal–1* or *mDsRed–Gal–1^S8A^* by electroporation and Gal–1 O–GlcNAcylation was analyzed as indicated in the Section 2. As shown in Figure 1C, the simulation of Aβ_1–42_ oligomers resulted in an increase in WGA-purified Gal–1, indicating an increased Gal–1 O–GlcNAcylation stoichiometry (Figure 1C, upper panel, lane 2). From a quantitative point of view, this stimulation resulted in a 2.01 ± 0.02-fold increase in O–GlcNAcylated Gal–1 (Figure 1C, column 2) compared to the control (Figure 1C, column 1). The addition of free GlcNAc demonstrated that WGA-agarose beads specifically purified O–GlcNAcylated proteins (Figure 1C, upper panel, lane and column 3).

Regarding the mutated Gal–1, our results indicated that the exchange of the Ser^8^ by Ala resulted in a strong decrease in the Gal–1 O–GlcNAcylation. Without Aβ_1–42_ oligomers stimulation, the O–GlcNAcylation levels were similar to the Gal–1^WT^ in non-stimulated cells (1 ± 0), and in the presence of Aβ_1–42_ oligomers, the levels were 1 ± 0. Thus, Aβ_1–42_ oligomers were not able to induce an increase in O–GlcNAcylation in Gal-1^S8A^. Altogether, our findings suggest that Aβ_1–42_ oligomers mediate the Ser^8^ O–GlcNAcylation of Gal–1 in human microglia.

Given that Aβ_1–42_ oligomers resulted in a rise in the microglial protein O–GlcNAcylation, with Gal–1 being among these proteins, our next goal was to investigate whether Aβ_1–42_ oligomers produced an increase in OGT enzymatic activity towards Gal–1. This was measured by an in vitro OGT activity assay.

As shown in Figure 1D, Aβ_1–42_ oligomers mediated OGT activation, which was measured as an increase in the O–GlcNAcylation of its substrate Gal–1. For the quantification, O–GlcNAcylated Gal–1 (Figure 1D, upper panel) was normalized with the total Gal–1 (Figure 1D, middle panel). Western blot with an anti-Myc antibody confirmed the immunoprecipitation of the enzyme OGT (Figure 1D, lower panel). Quantitatively, in the presence of Aβ_1–42_ oligomers, OGT was two-fold as active as in the non-stimulated conditions. The quantification of the Western blot demonstrated a fold change of 2 ± 0 compared to the control (Figure 1D).

### 3.2. O-GlcNAc Homeostasis Controls Aβ_1–42_-Induced Microglial Migration

Taking into account that OGT mediated the O–GlcNAcylation of Gal–1, a protein implicated in cellular migration, we investigated whether O–GlcNAcylation regulates microglia migration. This was analyzed by a scratch wound healing assay using the OGT inhibitor OSMI–I [28] and the OGA inhibitor Thiamet G [29]. Briefly, 8 × 10^5^ cells were seeded in a 6-well plate per well 48 h prior to the experiment and were then serum-starved for 24 h. On the day of the experiment, 25 μM of OSMI–I and 10 μM of Thiamet G or DMSO (as a vehicle) were added overnight. Next, a scratch was performed in the cell monolayer, and 5 μM Aβ_1–42_ oligomers were added in the corresponding wells. As expected, after 24 h in the presence of Aβ_1–42_ oligomers, they produced a strong increase in microglia migration (Figure 2D) compared to time 0 (Figure 2B), which resulted in a 440 ± 30% increase in migration compared to the control (Figure 2M). This vehicle (DMSO) did not alter migration. Regarding the effect of OSMI–I, this compound reduced basal microglia migration (Figure 2G), and OSMI–I abolished the Aβ_1–42_ oligomers-dependent increase in microglial migration (Figure 2H compared to Figure 2D). The migration rate was 41 ± 9% in the presence of OSMI-I and, in the absence of Aβ_1–42_ oligomers (Figure 2M), the addition of amyloid beta did not affect the migration of OSMI-treated cells (41 ± 9%, Figure 2M). On the contrary, the OGA inhibitor Thiamet G provoked a substantial increase in microglia migration. In fact, when cells were not stimulated with the Aβ_1–42_ oligomers, migration reached 539 ± 53% (Figure 2K,M). This was similar when the Aβ_1–42_ oligmers was added to microglia, which resulted in an increase in migration by 550 ± 66% (Figure 2L,M).

In order to verify these results, the effect of O–GlcNAcylation on microglia migration was analyzed by a transwell assay. In total, 250,000 cells were seeded in the upper well of the transwell chamber, they were treated with OSMI–I, Thiamet G or DMSO for 1 h followed by 5 μM Aβ_1–42_ oligomers, and 24 h of microglia migration was examined. The Aβ_1–42_ oligomers increased microglia migration (5517 ± 45%) (Figure 3B) compared to the control cells (Figure 3A). Again, OSMI–I completely blocked microglial migration induced by Aβ_1–42_ oligomers. From a quantitative point of view, the migration rate was 108 ± 13% (Figure 3D,G) and 88 ± 11% (Figure 3C,G) with and without Aβ_1–42_ oligomers’ stimulation, respectively. As expected, Thiamet G provoked an uncontrolled migration, even in the absence of Aβ_1–42_ oligomers, where the migration rate was 5737 ± 65% (Figure 3E,F), and the presence of Aβ_1–42_ oligomers did not modify this rate 5665 ± 165% (Figure 3F,G). These findings suggest that protein O–GlcNAcylation could control microglia migration.

### 3.3. O-GlcNAc Homeostasis Controls Aβ_1–42_ Oligomers-Induced Microglial Migration through Galectin-1 Ser^8^ O-GlcNAcylation

The OGT inhibitor, OSMI–I, reduced global O–GlcNAcylation in cells and affected cell viability [26]. Taking into account that we had identified by LC/MS the Gal–1 ACGLVA**S**NLNLKPGECLR peptide susceptible to O–GlcNAcylated in Serine 8, we generated and purified the TAT–OGT binding domain-Gal–1 (TAT–OBD–Gal–1) peptide. This permeant peptide interfered in the natural interaction between the OGT and Gal–1, blocking the Gal–1 O–GlcNAcylation and downstream signaling pathway.

Briefly, immortalized human microglia (800,000 cells/well) were seeded in 6-well plates 48 h prior to the experiment. These cells were serum-starved for 24 h. Then, they were treated with 25 µg of either the TAT–EGFP control peptide or 25 µg of TAT–OBD–Gal–1 for one hour. A scratch was performed in the cell monolayer with 5 µM Aβ_1–42_ oligomers, which were added, and images were taken at 0 h and 24 h of migration. As shown in Figure 4, TAT–EGFP-treated cells responded to Aβ_1–42_ oligomers by increasing their migration to 1225 ± 16% (Figure 4D,I) compared to control, which was expressed as 100% (Figure 4C,I). On the contrary, TAT–OBD–Gal–1 treatment completely blocked microglia migration. In fact, the migration rate was 138 ± 15% without Aβ_1–42_ oligomers stimulation (Figure 4G,I), which was similarly maintained at 127 ± 41% with the Aβ_1–42_ oligomers (Figure 4H,I).

Next, a transwell assay was performed with TAT–EGFP and TAT–OBD–Gal–1 peptides. In this regard, serum-deprived immortalized human microglia were seeded in the upper well of the transwell chamber (250,000 cells per condition), where they were treated with 25 µg of either the TAT–EGFP control peptide or 25 µg of TAT–OBD–Gal–1 for one hour followed by 5 µM Aβ_1–42_ oligomers. Migration was analyzed 24 h later. As shown in Figure 5, TAT–EGFP-treated cells responded to Aβ_1–42_ oligomers by increasing their migration significantly to 732 ± 25% (Figure 5B,E) compared to non-stimulated cells (Figure 5A,E). On the contrary, TAT–OBD–Gal–1 completely blocked microglia migration. These microglia migrated 84 ± 14% without the stimulation of Aβ_1–42_ oligomers (Figure 5C,E) which was similarly maintained at 95 ± 8% with Aβ_1–42_ oligomers (Figure 5D,E). Altogether, these results suggested that Aβ_1–42_ oligomers could involve the tandem of OGT/Gal-1 in microglia migration.

In order to examine the role of Gal–1^S8^ O–GlcNAcylation in the control of microglia migration, human microglia were transfected with 10 μg of plasmids, encoding either *mDsRed–Gal–1^wt^* or *mDsRed–Gal–1^S8A^* (non-O–GlcNAc Gal–1) and 24 h later; these cells were serum-starved for 24 h. Then, a scratch was conducted in the cell monolayer, where microglia were either treated or not with 5 μM Aβ_1–42_ oligomers, and migration was analyzed by taking images at 0 h and 24 h. As shown in Figure 6, microglia that had been transfected with *Gal–1^wt^* responded to Aβ_1–42_ oligomers (169 ± 3%) (Figure 6D,I) compared to the control (Figure 6C,I). *Gal-1^S8A^*-transfected cells, however, resulted in an uncontrolled migration both in the absence (194 ± 5%, Figure 6G,I) and in the presence of Aβ_1–42_ oligomers (193 ± 1%, Figure 6H,I). When we performed the microglia migration experiment in a transwell assay, the migration pattern was similar to the wound healing assay. Microglia that had been transfected with *Gal–1^WT^* responded to Aβ_1–42_ oligomers by migrating 1177 ± 136% (Figure 7B,E) compared to the control (Figure 7A,E). On the other hand, *Gal-1^S8A^*-transfected cells presented an uncontrolled migration both in the absence (1936 ± 156%, Figure 7C,E) and in the presence of Aβ_1–42_ oligomers (2274 ± 156%, Figure 7D,E).

Hence, our findings suggest that the Gal–1 Ser^8^O–GlcNAcylation fine-tuned microglial migration, and abolishing Gal–1 Ser^8^ O–GlcNAcylation resulted in a hyper-migration phenotype in microglia, which was indicative of uncontrolled microglial migration. Overall, our data suggest that Aβ_1–42_ oligomers regulated microglial migration through the modulation of Gal–1 Ser^8^ O–GlcNAcylation.

## 4. Discussion

The protein O–GlcNAcylation pattern is altered in the brains of AD patients, affecting myriad proteins, including synaptic, cytoskeleton and memory-associated proteins, all of which are relevant in the context of AD [6,8]. Accordingly, our study on microglia has revealed that Aβ_1–42_ oligomers provoke an increase in protein O–GlcNAcylation, and of the 55 proteins identified, we found Gal-1. In addition, LC/MS analysis suggested that the Gal–1 O–GlcNAcylation does occur on serine 8, verifying it both through biochemical and genetic approaches. The fact that Gal-1 is modified by the GlcNAc group points out that it could be an important PTM for cell biology [30,31,32,33]. In fact, we have demonstrated that this O–GlcNAcylation is relevant for migration in human microglia. Furthermore, we identified that the residue being modified is the Ser^8^ from Gal-1. As a matter of fact, it is interesting to highlight that the Gal-1 was reported as phosphorylated at that same Ser 8 residue [34,35]. Taking into account that phosphorylation and O–GlcNAcylation could compete for the same residues, a disequilibrium between both PTMs could provoke the aberrant function of Gal–1 in microglia, in our case, in the context of AD. Aβ_1–42_ oligomers could favor the balance towards O–GlcNAcylation as a detriment to phosphorylation. Hence, it would be interesting to investigate if AD patients present reduced phosphorylation, together with increased O–GlcNAcylation, in Gal–1. It would also be important to study whether this imbalance depends on the stage of the disease. We postulate that this imbalance towards Gal–1 O–GlcNAcylation could be happening at the initial stages, where microglia migration is increased. Thus, restoring the O–GlcNAcylation/phosphorylation equilibrium could be a therapeutic strategy in pathologies where this equilibrium is altered. In this regard, some investigations have conducted attempts with OGA inhibitors in order to increase tau protein O–GlcNAcylation and decrease phosphorylation [29,36]. Nevertheless, we believe that this approach should be specific to a particular protein. In fact, not all proteins present an equilibrium that is displaced towards O–GlcNAcylations in AD, as some of them are displaced towards phosphorylation [7,8,29,36,37].

Examining the functionality of O–GlcNAcylated Gal–1 induced by Aβ_1–42_ oligomers in the control of microglia migration, our findings suggest that O–GlcNAcylated Gal–1 maintains this response in a controlled manner. On the contrary, when we mutated serine 8 to alanine, which prevented O–GlcNAcylation at this site, although we expected to block the microglia migration induced by Aβ_1–42_ oligomers, this mutation led to a boost in the migration in an uncontrolled manner. This Gal–1 mutant behaved as if it was a constitutively active form. This increased migration could seem contradictory if we compare it with the inhibition of migration observed with the OGT inhibitor OSMI–I or the TAT–OBD–Gal–1 interfering peptide. The fact that OGT has a wide variety of substrates means that inhibiting its activity or using an interfering peptide could block the O–GlcNAcylation of many proteins rather than only Gal–1. Therefore, the response is different than when we specifically targeted the O–GlcNAcylation of Gal–1. Thus, in this paper, we have also described a peptide that could be blocking OGT activity.

Gal–1 is a protein that can function as a monomer or as a homodimer [30]. Its dimerization is essential to activating signaling cascades such as Ras/MAPK in the intracellular space [38] or CD45 phosphatase in the extracellular space [39]. We speculate that the presence of the GlcNAc group in Gal–1 favors its monomeric form over the dimeric form, while the S8A mutant facilitates Gal-1 homodimerization. Many functions of Gal–1 as a homodimer are related to its ability to function as a lectin by binding to glycoproteins of the cell membrane and components of the extracellular matrix [26], controlling cell responses such as proliferation, survival and inflammation in cancer or cell migration [30,40]. In the context of the central nervous system (CNS), it has been reported that astrocytes secrete Gal–1, which binds to CD45 glycoprotein in the microglia, to prevent the development of experimental autoimmune encephalomyelitis [39]. Nevertheless, these authors did not investigate if the microglia itself were able to secrete the Gal–1 protein and regulate its activation state through an autocrine mechanism. Future research needs to clarify whether O–GlcNAcylated Gal–1 is secreted and, secondly, whether it produces an autocrine/paracrine regulation in the microglia and other cell types in the CNS.

## 5. Conclusions

This study has revealed that in human microglia, Aβ_1–42_ oligomers induce an increase in the protein O–GlcNAcylation and OGT activity towards Gal-1. Moreover, we identified that one of these proteins is Gal–1. The O–GlcNAcylation of Gal–1 in Ser^8^ is key for Aβ_1–42_ oligomers-induced microglia migration. In addition, we generated a Gal–1 (S8A) that behaved as a constitutively active form of Gal–1, absolutely dysregulating microglia migration. Given the importance of microglial migration in AD, we do not discard that Gal–1 could play an important role during the progression of AD, especially at the initial stages where microglia are recruited to the amyloid plaques.

## Figures and Tables

**Figure 1 cells-12-01876-f001:**
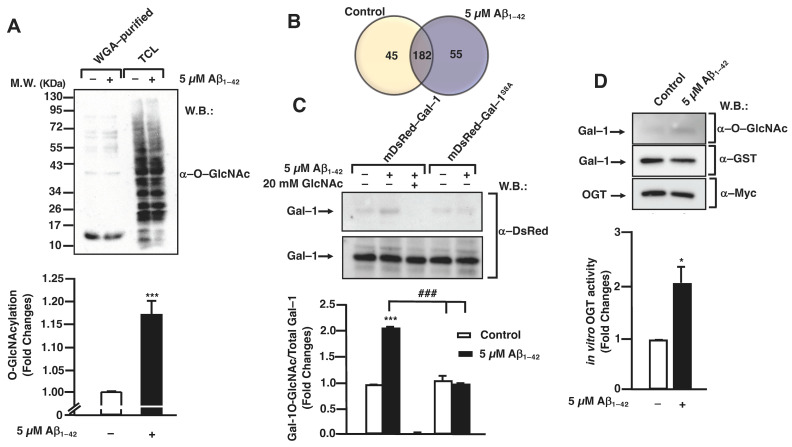
Aβ_1–42_ oligomers-stimulated posttranslational modifications by O-GlcNAcylation in microglia. (**A**) Microglia were stimulated with 5 μM Aβ_1–42_ oligomers for 30 min, lysed in RIPA buffer and O–GlcNAcylated proteins were purified by WGA-agarose beads. Purified and total input fractions were analyzed by SDS-PAGE and Western blot with an antibody against O–GlcNAc. The bands used for quantification were those between molecular weights of 55 and 72 KDa. The blot was representative of three independent experiments, and the bar graph represents the mean ± S.E.M. of three independent experiments (N = 3) compared to the control as fold change. Mann–Whitney statistical analysis showed a significant difference. *** *p* < 0.001. (**B**) Microglia were either stimulated or not with 5 µM Aβ_1–42_ oligomers for 30 min, and O–GlcNAcylated proteins were purified with WGA-agarose beads. The proteins present in each group were identified by LC-MS/MS. Numbers inside the Venn diagram indicate the number of proteins that were identified exclusively in each group and the number of proteins that were common in both groups. The Venn diagram was representative of three independent experiments, with the difference in proteins identified between them being ±5. (**C**) Microglia were transfected either with *mDsRed–Gal–1* or *mDsRed-Gal–1^S8A^* and were stimulated with 5 µM Aβ_1–42_ oligomers, and O–GlcNAcylated proteins were purified with WGA-agarose beads. Free GlcNAc was added as a control for specificity in the purification. Purified and total input fractions were analyzed by SDS-PAGE and Western blot. The blot was representative of three independent experiments (N = 3), and the bar graph represents the mean ± S.E.M. of each WGA-purified condition, normalized by their corresponding total input, compared to the non-stimulated control. Mann–Whitney statistical analysis showed a significant difference. *** *p* < 0.001; ### *p* < 0.001. (**D**) *Myc–OGT* was transfected in microglia, where they were stimulated with 5 µM Aβ_1–42_ oligomers, and immunoprecipitation was performed with α-myc. Substrates UDP–GlcNAc and GST-Gal-1 were added to the complexes, and the O-GlcNAcylation reaction was incubated. OGT activity was visualized by SDS-PAGE and Western blot. The blot was representative of three independent experiments (N = 3). An antibody α–O–GlcNAc was used for analyzing OGT activity; an antibody α-GST was used in order to check the total substrate; an antibody α-myc was used in order to confirm the immunoprecipitation efficiency. O–GlcNAcylated Gal–1 was normalized by its corresponding total substrate (α–GST) in each condition. These normalized values were compared to the non-stimulated control (fold change). The bar graph represents the fold change means ± S.E.M. of each condition. Mann–Whitney statistical analysis showed a significant difference * *p* < 0.05.

**Figure 2 cells-12-01876-f002:**
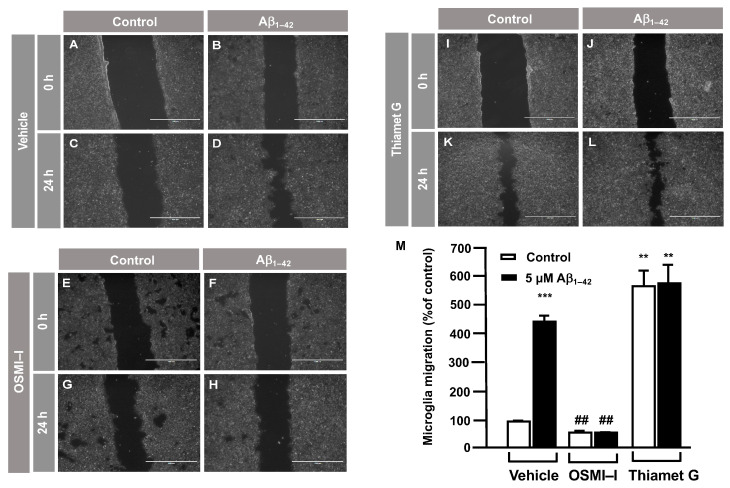
Protein O–GlcNAcylation mediated by Aβ_1–42_ oligomers control microglia migration: an analysis of cell migration by an in vitro wound healing assay. Microglia were treated with either a vehicle (DMSO), OSMI–I or Thiamet G, and they were either stimulated or not with 5 µM Aβ_1–42_ oligomers. Cell migration was analyzed after 24 h of incubation. (**A**–**L**). Representative photographs of a randomly chosen field of one independent experiment out of three. The scale bar in the lower part of each photograph is 1000 μm. (**M**) The bar graph represents the quantification of the migration. Values indicate the mean ± S.E.M. of the migrated distance (expressed as the % of control) in four randomly chosen fields of three independent experiments (N = 3. Mann–Whitney statistical analysis showed a significant difference. ## *p* < 0.01; ** *p* < 0.01; *** *p* < 0.001.

**Figure 3 cells-12-01876-f003:**
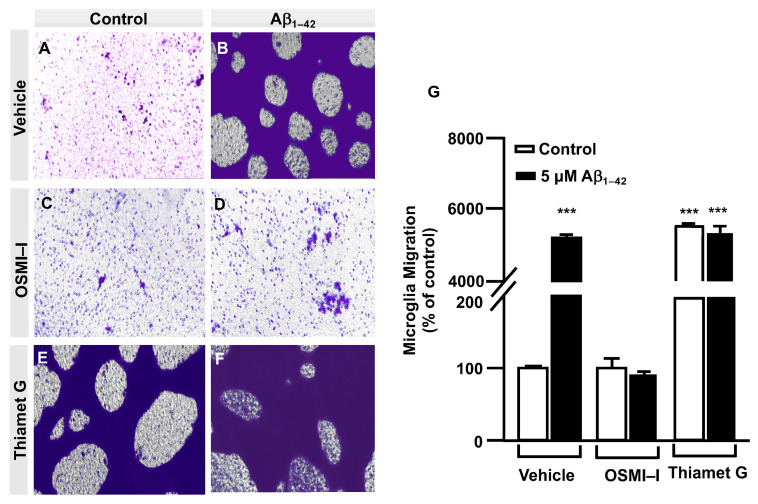
Protein O–GlcNAcylation mediated by Aβ_1–42_ oligomers control microglia migration: Quantitative and qualitative analysis of microglia migration assessed by an in vitro transwell assay. Microglia were treated with either a vehicle (DMSO), OSMI–I or Thiamet G and were either stimulated or not with 5 µM Aβ_1–42_ oligomers. Cell migration was analyzed after 24 h of incubation. (**A**–**F**) Representative photographs of a randomly chosen field of one independent experiment out of three. Purple areas correspond to all the cells that migrated, as stained with Mayer’s hematoxylin. (**G**) The bar graph represents the quantification of the migration. Values indicate the mean ± S.E.M. of the stained area (corresponding to the migrated cells) expressed as the % of control in four randomly chosen fields of three independent experiments (N = 3). Mann–Whitney statistical analysis showed a significant difference. *** *p* < 0.001.

**Figure 4 cells-12-01876-f004:**
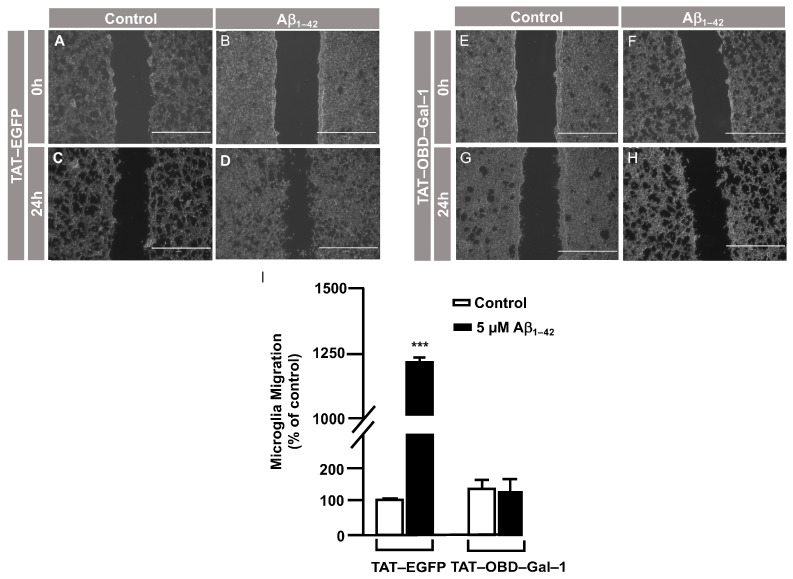
TAT–OBD–Gal–1 peptide blocks Aβ_1–42_ oligomer-induced microglia migration: analysis of cell migration by in vitro wound healing assay. Microglia were treated with either TAT–EGFP or TAT–OBD–Gal–1 and were either stimulated or not with 5 µM Aβ_1–42_ oligomers. Cell migration was analyzed after 24 h of incubation. (**A**–**H**). Representative photographs of a randomly chosen field of one independent experiment out of three. The scale bar in the lower part of each photograph is 1000 μm. (**I**) The bar graph represents the quantification of the migration. Values indicate the mean ± S.E.M. of the migrated distance (expressed as the % of control) in four randomly chosen fields of three independent experiments (N = 3). Mann–Whitney statistical analysis showed a significant difference. *** *p* < 0.001.

**Figure 5 cells-12-01876-f005:**
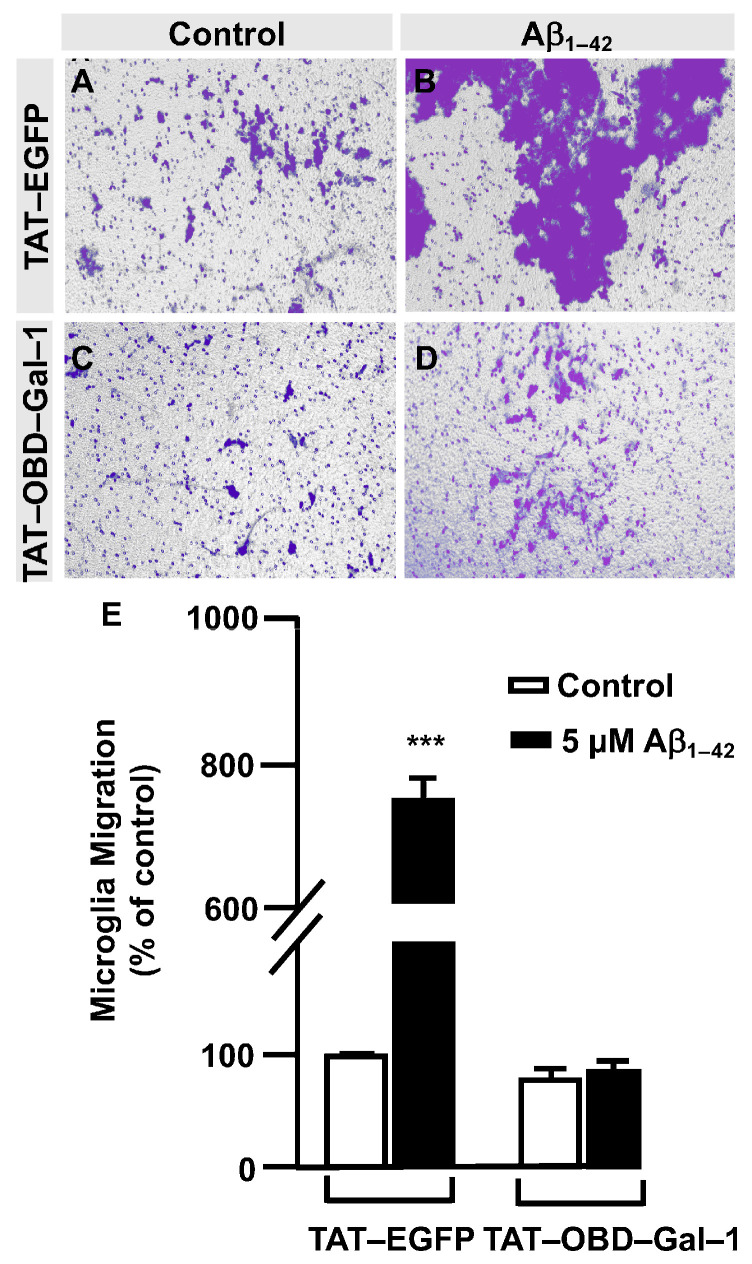
TAT–OBD–Gal–1 peptide blocks Aβ_1–42_ oligomers-induced microglia migration: quantitative and qualitative analysis of microglia migration assessed by an in vitro transwell assay. Microglia were treated with either TAT–EGFP or TAT–OBD–Gal–1 and were either stimulated or not with 5 µM Aβ_1–42_ oligomers. Cell migration was analyzed after 24 h of incubation. (**A**–**D**). Representative photographs of a randomly chosen field of one independent experiment out of three. (**E**). The bar graph represents the quantification of the migration. Values indicate the mean ± S.E.M. of the stained area (expressed as the % of control) in four randomly chosen fields of three independent experiments (N = 3). Mann–Whitney statistical analysis showed a significant difference. *** *p* < 0.001.

**Figure 6 cells-12-01876-f006:**
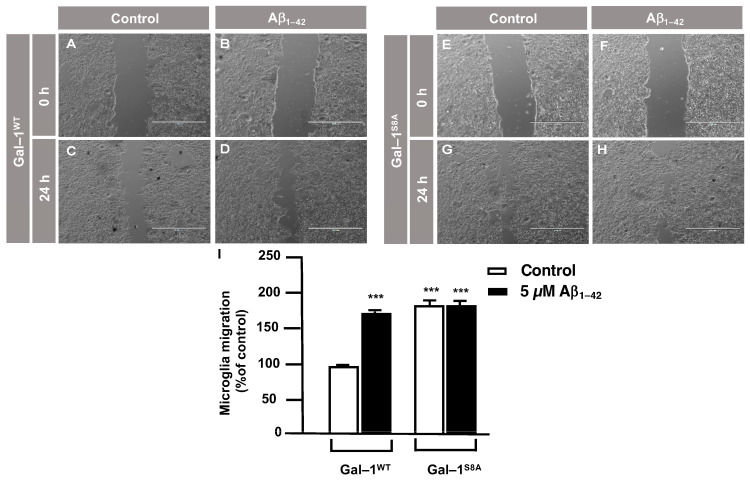
Galectin–1 O–GlcNAcylation regulates microglia migration in a controlled manner: analysis of cell migration by an in vitro wound healing assay. Microglia were transfected with 10 µg of either *mDsRed–Gal–1^WT^* or *mDsRed–Gal–1^S8A^*, which were either stimulated or not with 5 µM Aβ_1–42_ oligomers. Cell migration was analyzed after 24 h of incubation. (**A**–**H**). Representative photographs of a randomly chosen field of one independent experiment out of three. The scale bar in the lower part of each photograph is 1000 μm. (**I**). The bar graph represents the quantification of the migration. Values indicate the mean ± S.E.M. of the migrated distance (expressed as the % of control) in four randomly chosen fields of three independent experiments (N = 3). Mann–Whitney statistical analysis showed a significant difference. *** *p* < 0.001.

**Figure 7 cells-12-01876-f007:**
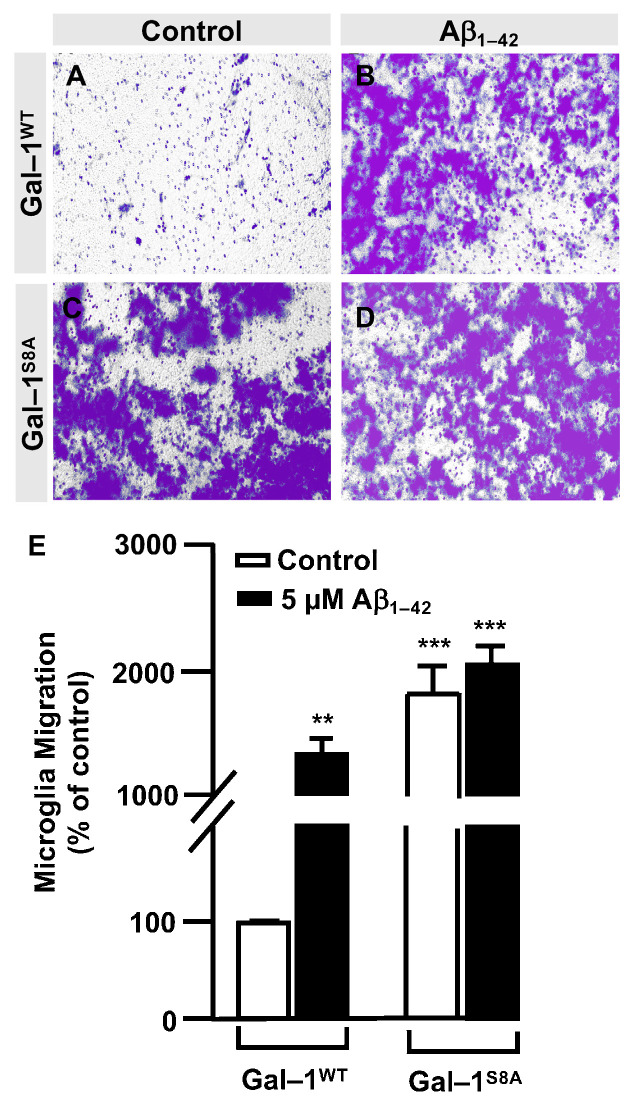
Galectin–1 O–GlcNAcylation regulates microglia migration in a controlled manner: quantitative and qualitative analysis of melanoma cell migration assessed by an in vitro transwell assay. Microglia cells were transfected with 10 µg of either *mDsRed–Gal–1^WT^* or *mDsRed–Gal–1^S8A^* and were either stimulated or not with 5 µM Aβ_1–42_ oligomers. Cell migration was analyzed after 24 h of incubation. (**A**–**D**). Representative photographs of a randomly chosen field of one independent experiment out of three. (**E**). The bar graph represents the quantification of the migration. Values indicate the mean ± S.E.M. of the stained area (expressed as the % of control) in four randomly chosen fields of three independent experiments (N = 3). Mann–Whitney statistical analysis showed a significant difference. ** *p* < 0.01; *** *p* < 0.001.

## Data Availability

Not applicable.

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
