# Peer review of "Amyloid β1–42 Oligomers Induce Galectin–1S8 O–GlcNAcylation Leading to Microglia Migration"

_cells, 2023, doi:10.3390/cells12141876_

Round 1

Reviewer 1 Report

In this article, Arrazola Sastre et al. used Amyloid b1-42 peptide, human immortalized microglia cells, and biochemical with genetic approaches and showed that the Abeta1-42 peptide stimulated the activity of O-linked GlcNAc Transferanse, enhanced O-GlcNacylation on Gal-1 which lead microglia migration. It is an interesting finding, but there are many concerns regarding this paper.

The authors showed that the O-GlcNAcylated Gal-1 induced microglia migration, but I could not see any description or experiment indicating a possible mechanism to control these cellular responses through Gal-1 Ser8 O-GlcNAcylation.

The immunoblot result for Gal-1 was not convincing to me, and the band intensity was really weak, especially in Figure 1C upper and 1D upper. I felt that the bar graph in Figure 1A is exaggerated. The authors used 5 uM peptides and 30 minutes of incubation for microglial stimulation. The author should perform the dosing experiment, determine the best concentration and stimulation time, and show it in the result section. Which bands were used for quantification in Figure 1A?

The authors showed that Abeta1-42 peptides increased O-GlcNacylation on Gal-1, which resulted in microglia migration. Is this phenomenon specific to the Abeta1-42 peptide? The authors should test if other peptides affect microglial migration as a control condition.

The material and method section should be written in detail to be repeated by others.
The material and method section should be written in detail in order to be repeated by other people. If you look at other Cells research articles, you'll see the reagent company name, headquarters, and country in the material and method section. The authors did not provide this information, but this is the standard format for this journal, so the same should be followed in your manuscript. The component of buffers and solutions should be described.
What is 2-mercaptoethanol for in line 74?
No antibody information, anti-DsRed antibody in line 141, and specific antibodies in line 152. What is the dilution of antibody and incubation time, and what is the blocking buffer?
How much proteins are loaded for immunoblot and purification?
What is the loading buffer in line 150?
How to transfer the protein, and what membrane is used?
The abbreviation (OGT) was used without definition.

Line 193, A histogram is used to represent the frequency distribution of a few data points of one variable. It is a bar graph.
Line 287, Figure # missed
Line 312, no “M” in Figure 3
No scale bar in Figures 3, 5, and 7.
Line 78, 83, and 115, what does the underline mean?

Author Response

REVIEWER 1:

In this article, Arrazola Sastre et al. used Amyloid b1-42 peptide, human immortalized microglia cells, and biochemical with genetic approaches and showed that the Abeta1-42 peptide stimulated the activity of O-linked GlcNAc Transferanse, enhanced O-GlcNacylation on Gal-1 which lead microglia migration. It is an interesting finding, but there are many concerns regarding this paper:

Point 1

1) The authors showed that the O-GlcNAcylated Gal-1 induced microglia migration, but I could not see any description or experiment indicating a possible mechanism to control these cellular responses through Gal-1 Ser8 O-GlcNAcylation. 

Response: We agree that we did not address the possible mechanisms to control microglia migration though Gal-1. Actually, when we saw through MS that Gal-1 was likely to undergo post-translational modifications by O-GlcNAcylation induced by Aß and knowing that this family of proteins are key elements in cancer progression, the question we asked ourselves was: What role could Gal-1 be playing in microglia biology, and even more so and that serine 8?. The first thing we did was to investigate the relationship between Aß and Gal-1 with migration and we saw that this axis was indeed related to migration, but more importantly, the Gal-1serine 8 O-GlcNAcylation was key for orderly migration to occur, since the non-O- O-GlcNAcylable Gal-1 lost that control and the migration became aberrant. These results seem relevant to us to communicate to the scientific community, although we reiterate that more research is needed to reveal the mechanisms that regulate Gal-1 both upstream and downstream. In fact, we have preliminary results that suggest that GTPase Rac1 and glycogen phosphorylase upstream regulate the Gal-1 O-GlcNAcylation.

Point 2

1) The immunoblot result for Gal-1 was not convincing to me, and the band intensity was really weak, especially in Figure 1C upper and 1D upper.

Response: We think that despite the weak signal of the blots, the results are clear, especially since the controls used indicate so. It is true that we could have overexposed, but we decided for this option because if we overexposed it would be lost in subtlety. It must also be taken into account that the purification of O- GlcNAcylation proteins by WGA is not always very satisfactory in an experimental point of view.

2) I felt that the bar graph in Figure 1A is exaggerated.

Response: In our opinion it is more visual, and the result itself does not change.

3) The authors used 5 uM peptides and 30 minutes of incubation for microglial stimulation. The author should perform the dosing experiment, determine the best concentration and stimulation time, and show it in the result section.

Response: We had already established both the time (30 min) and the Aβ concentration (5µM) as optimal (Manterola, L. et al. Transl Psychiatry (2013) 3, e219; doi:10.1038/tp.2012.147; Wyssenbach, A et al. Aging Cell 2016 Dec;15(6):1140-1152.) to investigate early intracellular signaling in both neurons and astrocytes, and on this basis we have continued, in this case with microglia, and it works very well too.

             4) Which bands were used for quantification in Figure 1A?

Response: The bands between 72 and 55 Kd of lines 1 and 2.

 Point 3

The authors showed that Abeta1-42 peptides increased O-GlcNacylation on Gal-1, which resulted in microglia migration. Is this phenomenon specific to the Abeta1-42 peptide? The authors should test if other peptides affect microglial migration as a control condition.

Response: We are not aware of it, but we believe that it is not a specific response to treatment with Aß, we think that Gal-1 O-GlcNAcylation is required to control migration, so probably other trophic factors that induce cell migration use this O-GlcNAcylation of the Gal-1 for its control.

Point 4

1) The material and method section should be written in detail to be repeated by others. If you look at other Cells research articles, you'll see the reagent company name, headquarters, and country in the material and method section. The authors did not provide this information, but this is the standard format for this journal, so the same should be followed in your manuscript.

Response: In the new version we provide this information.

2) The component of buffers and solutions should be described.

Response: In the new version we provide this information.

4) What is 2-mercaptoethanol for in line 74?.

Response: It was a mistake, we have corrected it.

5) No antibody information, anti-DsRed antibody in line 141, and specific antibodies in line 152. What is the dilution of antibody and incubation time, and what is the blocking buffer?.

Response: In the new version we provide this information. The blocking buffer is a well-established standard buffer that can be purchased from Merck or Thermo Science that is used to block non-specific sites on blotting membranes. We think that it is not necessary to specify the components.

6) How much proteins are loaded for immunoblot and purification?.

Response: We indicate in materials and methods section the number of cells with which we start, as well as the amount of protein that we have in the lysates and that we are going to immunoprecipitate or do WGA chromatography.

7) What is the loading buffer in line 150?

Response: Loading buffer also known Laemli’s buffer is a very well and standard buffer. In the new version we provide this information.

9) How to transfer the protein, and what membrane is used?

Response: The methodology that involves western blot is a very well established and standardized technique, so we do not believe that it is necessary to describe the technique in the manuscript.

10) The abbreviation (OGT) was used without definition.

Response: We have explained it in the new version

11) Line 193, A histogram is used to represent the frequency distribution of a few data points of one variable. It is a bar graph.

Response: We agree. we have corrected it.

12) Line 287, Figure # missed

Response: We agree. we have corrected it.

13) Line 312, no “M” in Figure 3

Response: We agree. we have corrected it.

14) No scale bar in Figures 3, 5, and 7.

Response: We agree. we have corrected it.

15) Line 78, 83, and 115, what does the underline mean?

Response: It was a mistake, we have corrected it.

Reviewer 2 Report

The manuscript by Arrazola Sastre et al. describes a series of cell biological experiments designed to demonstrate that amyloid Abeta1-42 peptide induces O-GlcNAcylation of galectin-1 on residue Ser8, which in turn is proposed to affect microglia migration in Alzheimers disease. Galectin-1 was identified as one of 55 proteins pulled down by a wheat germ agglutinin affinity colum after stimulation of microglia with Abeta1-42.

The experiments are in general convincing and are well-presented. The only weakness from my point of view is that the existence of an O-GlcNAcylated species is never directly proven by e.g. MS, but rather inferred from a series of indirect experiments. On lines 204-205 it is written that "we identified the ACGLVASNLNLKPGECLR peptide". Surely it should have been possible to observe the glycosylated species directly in this experiment?

Some comment on the relationship of the detected peptide to proposed consensus sequences for O-GlcNAcylation might be in order.

Lines 231 and 236: It is written that the peptide provoked a rise in OGT activity. Surely what it does is provoke a rise in the amount of OGT produced? Or is the peptide proposed to have a stimulatory effect on the OGT enzyme itself through a peptide-protein interaction? This could be clarified at some point in the manuscript.

Sometimes changes in some activity are expressed as absolute numbers, sometimes as percentages. Perhaps this should be standardised throughout. Also, I question whether two decimal places are necessary when quoting percentage changes in the thousands with uncertainties in the hundreds. I would use only one decimal place, perhaps none.

Some suggestions for linguistic improvement:

Line 27: "modifications" should be "modification"
Line 51: "of these cells" could be "examples of such cells"
Line 56: "its" should be "their"
Line 63: "Galectin" should be "galectin"
Line 64: "on the Serine 8" should be "on serine 8"
Line 66: "these" should be "this"
Lines 78-82: why are some of the words underlined? Are they supposed to be hyperlinks? If so, why are there not more hyperlinks for such technical terms in the text?
lines 81-82: "isopropyl", "nickel" and "agarose" should not be capitalised
line 97: cells can only be deprived OF something. What were they deprived of?
line 98: "amyloid" should not be capitalised
line 109: "was" should be "were"
line 187: "wheat" should not be capitalised
line 243: Should it not read "Western blot with an antibody against..."?
line 246: "was" should be "were"
line 247:  add "and" between "minutes" and "O-GlcNAcylated"?
line 257: "and the histogram"
line 270: no comma between "account" and "that"
line 288: "increase of migration"
line 291: "control" should be "controls"
line 336: Why is the microscope specified only here? Wasn't the same microscope used for other cell migration figures? This is mentioned in Materials and Methods anyway so unnecessary here.
line 375: remove "for"
line 384: "similar to the wound healing assay"
line 417: "patterns are altered"
line 421: The experiments surely demonstrate that Gal-1 O-GlcNAcylation does occur on serine 8, not just that it could? This is stated on line 425. Lines 421-427 should be rewritten.
line 428: no comma between "account" and "that"
line 448: "as if it was a"

Author Response

REVIEWER 2:

The manuscript by Arrazola Sastre et al. describes a series of cell biological experiments designed to demonstrate that amyloid Abeta1-42 peptide induces O-GlcNAcylation of galectin-1 on residue Ser8, which in turn is proposed to affect microglia migration in Alzheimers disease. Galectin-1 was identified as one of 55 proteins pulled down by a wheat germ agglutinin affinity colum after stimulation of microglia with Abeta1-42.

Point 1

1) The experiments are in general convincing and are well-presented. The only weakness from my point of view is that the existence of an O-GlcNAcylated species is never directly proven by e.g. MS, but rather inferred from a series of indirect experiments. On lines 204-205 it is written that "we identified the ACGLVASNLNLKPGECLR peptide". Surely it should have been possible to observe the glycosylated species directly in this experiment?.

Response: No, using the approach we have made, it is not possible to directly see glycosylated species. What we have done is to separate by affinity chromatography using a WGA matrix, proteins that are found to be O-glycosylated and by means of MS the indicated ones have been identified. One of them was gal-1 and according to the MS result, one of the peptides identified was ACGLVASNLNLKPGECLR and it presented a potential O-glycosylation site that was serine 8. We later verified it by biochemical assays.

2) Some comment on the relationship of the detected peptide to proposed consensus sequences for O-GlcNAcylation might be in order.

Response: The O-glycosidic mechanism is not as complex as that of N-glycosylation. Proteins trafficked into the Golgi are most often O-glycosylated by O-N-acetylglucosamine transferase (OGT), which transfers a single GalNAc residue to the β-OH group of serine or threonine. To date, there is no known consensus sequence for this enzyme, although structural motifs have been characterized. Some proteins are O-glycosylated with GlcNAc, fucose, xylose, galactose or mannose, depending on the cell and species (Spiro RG (2002) Protein glycosylation: Nature, distribution, enzymatic formation, and disease implications of glycopeptide bonds. Glycobiology 12:43R–56R, Dell A, Morris HR (2001) Glycoprotein structure determination by mass spectrometry. Science 291:2351–6).

3) Lines 231 and 236: It is written that the peptide provoked a rise in OGT activity. Surely what it does is provoke a rise in the amount of OGT produced? Or is the peptide proposed to have a stimulatory effect on the OGT enzyme itself through a peptide-protein interaction? This could be clarified at some point in the manuscript.

Response: We agree. We have redrafted the text: Aβ1-42 oligomers mediated OGT activation, measured as an increase in the O-GlcNAcylation of its Gal-1 substrate.

4) Sometimes changes in some activity are expressed as absolute numbers, sometimes as percentages. Perhaps this should be standardised throughout. Also, I question whether two decimal places are necessary when quoting percentage changes in the thousands with uncertainties in the hundreds. I would use only one decimal place, perhaps none.

Response: Regarding the microglia migration, we have presented the results as a percentage over the control, whereas in the O-glycosylation changes we have preferred to represent them as fold change. It seemed to us the best way to differentiate the expression of the experiments. On the other hand, and following the reviewer's recommendation, we have eliminated all decimal places.

5) Some suggestions for linguistic improvement:

Line 27: "modifications" should be "modification"

Line 51: "of these cells" could be "examples of such cells"

Line 56: "its" should be "their"

Line 63: "Galectin" should be "galectin"

Line 64: "on the Serine 8" should be "on serine 8"

Line 66: "these" should be "this"

Lines 78-82: why are some of the words underlined? Are they supposed to be hyperlinks? If so, why are there not more hyperlinks for such technical terms in the text?

Lines 81-82: "isopropyl", "nickel" and "agarose" should not be capitalized

Line 97: cells can only be deprived OF something. What were they deprived of?

Line 98: "amyloid" should not be capitalized

Line 109: "was" should be "were"

Line 187: "wheat" should not be capitalized

Line 243: Should it not read "Western blot with an antibody against..."?

Line 246: "was" should be "were"

Line 247:  add "and" between "minutes" and "O-GlcNAcylated"?

Line 257: "and the histogram"

Line 270: no comma between "account" and "that"

Line 288: "increase of migration"

Line 291: "control" should be "controls"

Line 336: Why is the microscope specified only here? Wasn't the same microscope used for other cell migration figures? This is mentioned in Materials and Methods anyway so unnecessary here.

Line 375: remove "for"

Line 384: "similar to the wound healing assay"

Line 417: "patterns are altered"

Line 421: The experiments surely demonstrate that Gal-1 O-GlcNAcylation does occur on serine 8, not just that it could? This is stated on line 425. Lines 421-427 should be rewritten.

Line 428: no comma between "account" and "that"

Line 448: "as if it was a"

Response: We thank the reviewer, we have accepted all of them

Reviewer 3 Report

The study by Sastre et. Al is an interesting investigation on the effects of the Aβ1-42 peptide on Protein O-GlcNAcylation and microglia cells. The study is of potential interest for the community. Before considering it for publication the following points need to be addressed.

Comments:

    • references need to be added at line 38, 45, 46,49, 53, 54, 203

    • line 49 is a strong statement

    • in the M&M section, with the exception of  the Abeta purification protocol, no references to previous works either from the Authors or other studies are reported. Are all these protocols completely new protocols? If not, references should be added for completion

    • The whole study has been carried out using Abeta peptides instead of aggregated forms, which are relevant species in the AD pathology. Do the authors have data concerning the effects of Abeta aggregates / oligomers on microglia activation / O-GlcNAcylation or do they plan to do this in the future? Either additional data or an expanded discussion on this point should be added in order to make the study complete

Minor comment:

line 19 “in addition...in turn” and line 209 “next, we first” are repetitions

Author Response

REVIEWER 3:

The study by Sastre et. Al is an interesting investigation on the effects of the Aβ1-42 peptide on Protein O-GlcNAcylation and microglia cells. The study is of potential interest for the community. Before considering it for publication the following points need to be addressed.

1) References need to be added at line 38, 45, 46,49, 53, 54, 203

Response: We agree. In the new version we add those references.

2) line 49 is a strong statement.

Response: We have included the following reference: Park, J.; Ha, H. J.; Chung, E. S.; Baek, S. H.; Cho, Y.; et al. O-GlcNAcylation ameliorates the pathological manifestations of Alzheimer's disease by inhibiting necroptosis. Sci Adv. 2021,7

3) in the M&M section, with the exception of the Abeta purification protocol, no references to previous works either from the Authors or other studies are reported. Are all these protocols completely new protocols? If not, references should be added for completion.

Response: We agree. In the new version we add those references.

4) The whole study has been carried out using Abeta peptides instead of aggregated forms, which are relevant species in the AD pathology. Do the authors have data concerning the effects of Abeta aggregates / oligomers on microglia activation / O-GlcNAcylation or do they plan to do this in the future? Either additional data or an expanded discussion on this point should be added in order to make the study complete.

Response: We agree. Actually, the recombinant Aß that we produce when we resuspend it, we configure the oligomeric and work with the oligomeric forms. This Figure shows the oligomeric forms we used.

10

17

26

34

43

Monomeric Aβ1-42

Oligomeric Aβ1-42

1    2

Mw (kDa)

Figure. Aβ1-42 species characterization. First lane shows 10 µL of commercial Aβ1-42 (Bachem); the second one shows 5µL recombinant Aβ1-42 produced in our laboratory. Aβ1-42 was separated by SDS-PAGE followed by Western blot. Immunoreactive bands using anti-β-Amyloid (clone 6E10) antibody were visualized by ECL.

What happened in the first version of the manuscript was that in the methodology we did not explain how we produced it, we referenced the article we followed for it. And as a consequence of this when in the text we indicated Aß peptide and indeed it is not correct. In the new version, we have added the complete methodology for obtaining Aß and we have called Aß oligomers instead of Aß peptide.

Round 2

Reviewer 1 Report

Point 1
1) The authors showed that the O-GlcNAcylated Gal-1 induced microglia migration, but I could not see any description or experiment indicating a possible mechanism to control these cellular responses through Gal-1 Ser8 O-GlcNAcylation.
Response: We agree that we did not address the possible mechanisms to control microglia migration though Gal-1. Actually, when we saw through MS that Gal-1 was likely to undergo post-translational modifications by O-GlcNAcylation induced by Aß and knowing that this family of proteins are key elements in cancer progression, the question we asked ourselves was: What role could Gal-1 be playing in microglia biology, and even more so and that serine 8?. The first thing we did was to investigate the relationship between Aß and Gal-1 with migration and we saw that this axis was indeed related to migration, but more importantly, the Gal-1serine 8 O-GlcNAcylation was key for orderly migration to occur, since the non-O- O-GlcNAcylable Gal-1 lost that control and the migration became aberrant. These results seem relevant to us to communicate to the scientific community, although we reiterate that more research is needed to reveal the mechanisms that regulate Gal-1 both upstream and downstream. In fact, we have preliminary results that suggest that GTPase Rac1 and glycogen phosphorylase upstream regulate the Gal-1 O-GlcNAcylation.

When I reviewed the manuscript, I felt that it was interesting findings. However, I could not see any possible mechanisms, and the authors agreed that they did not address the possible mechanisms to control microglia migration though Gal-1. As Cells have a high impact factor journal (Impact Factor: 7.666), the manuscript without mechanisms is not sufficient level. If the Aβ-binding receptors on microglia are involved in glycosylation and this migration, it would add more value.

Point 2
3) The authors used 5 uM peptides and 30 minutes of incubation for microglial stimulation. The author should perform the dosing experiment, determine the best concentration and stimulation time, and show it in the result section.
Response: We had already established both the time (30 min) and the Aβ concentration (5µM) as optimal (Manterola, L. et al. Transl Psychiatry (2013) 3, e219; doi:10.1038/tp.2012.147; Wyssenbach, A et al. Aging Cell 2016 Dec;15(6):1140-1152.) to investigate early intracellular signaling in both neurons and astrocytes, and on this basis we have continued, in this case with microglia, and it works very well too.

Authors should add these references in the manuscript.

4) Which bands were used for quantification in Figure 1A?
Response: The bands between 72 and 55 Kd of lines 1 and 2.

The authors should describe this in the manuscript.

Point 3
The authors showed that Abeta1-42 peptides increased O-GlcNacylation on Gal-1, which resulted in microglia migration. Is this phenomenon specific to the Abeta1-42 peptide? The authors should test if other peptides affect microglial migration as a control condition.
Response: We are not aware of it, but we believe that it is not a specific response to treatment with Aß, we think that Gal-1 O-GlcNAcylation is required to control migration, so probably other trophic factors that induce cell migration use this O-GlcNAcylation of the Gal-1 for its control.
If the migration of microglia is not specific to the Abeta1-42 peptide, this manuscript would be misleading.

Point 4
5) No antibody information, anti-DsRed antibody in line 141, and specific antibodies in line 152. What is the dilution of antibody and incubation time, and what is the blocking buffer?
Response: In the new version we provide this information. The blocking buffer is a well-established standard buffer that can be purchased from Merck or Thermo Science that is used to block non-specific sites on blotting membranes. We think that it is not necessary to specify the components.
9) How to transfer the protein, and what membrane is used?
Response: The methodology that involves western blot is a very well established and standardized technique, so we do not believe that it is necessary to describe the technique in the manuscript.

Authors responded that we think ..., we do not believe…, it is not the point. Authors should follow the instructions for authors. According to Materials and Methods in "instructions for authors": They should be described with sufficient detail to allow others to replicate and build on published results. New methods and protocols should be described in detail while well-established methods can be briefly described and appropriately cited. Give the name and version of any software used and make clear whether computer code used is available. Include any pre-registration codes.

Author Response

COMMENTS TO REVIEWER

Point  1

1) The authors showed that the O-GlcNAcylated Gal-1 induced microglia migration, but I could not see any description or experiment indicating a possible mechanism to control these cellular responses through Gal-1 Ser8 O-GlcNAcylation.
Response: We agree that we did not address the possible mechanisms to control microglia migration though Gal-1. Actually, when we saw through MS that Gal-1 was likely to undergo post-translational modifications by O-GlcNAcylation induced by Aß and knowing that this family of proteins are key elements in cancer progression, the question we asked ourselves was: What role could Gal-1 be playing in microglia biology, and even more so and that serine 8?. The first thing we did was to investigate the relationship between Aß and Gal-1 with migration and we saw that this axis was indeed related to migration, but more importantly, the Gal-1serine 8 O-GlcNAcylation was key for orderly migration to occur, since the non-O- O-GlcNAcylable Gal-1 lost that control and the migration became aberrant. These results seem relevant to us to communicate to the scientific community, although we reiterate that more research is needed to reveal the mechanisms that regulate Gal-1 both upstream and downstream. In fact, we have preliminary results that suggest that GTPase Rac1 and glycogen phosphorylase upstream regulate the Gal-1 O-GlcNAcylation.

1- When I reviewed the manuscript, I felt that it was interesting findings. However, I could not see any possible mechanisms, and the authors agreed that they did not address the possible mechanisms to control microglia migration though Gal-1. As Cells have a high impact factor journal (Impact Factor: 7.666), the manuscript without mechanisms is not sufficient level. If the Aβ-binding receptors on microglia are involved in glycosylation and this migration, it would add more value.

Response: Our apologies for the lack of clarity in our previous response. It is well-known that O-GlcNAcylation, the addition of O-GlcNAc sugar moieties to serine and threonine residues of proteins, can modulate various cellular processes by affecting protein function and signaling pathways. However, it is well known that in Alzheimer's disease the pattern of O-glycosylated proteins is altered with respect to health conditions. We wanted to know how the administration of Ab oligomers affected the protein O-glycosylation pattern in microglia. As we described, we heve been able to identify 55 proteins, one of which is galectin. The next step was to examine that the Ab oligomers induced O-glycosylation of Gal-1 protein, via the OGT enzyme, and it does. At this moment we had that the Ab oligomers induced the glycosylation of gal-1O- glycosylation by means of OGT. The next point was to investigate whether serine 8 was the serine that was o-glycosylated and we have shown this. Finally, with this mechanism, we try to understand what function or functions of microglia it was involved in, and we have shown that, once again, the Ab oligomers require OGT and in of galectin 1 Serine 8 O-glycosylation to control microglia. This is the first time that the relationship between Ab oligomers and microglia migration through the OGT and requires Gal-1 O-glycosylation at serine 8 has been demonstrated mechanistically. This is the novel intracellular signaling mechanism that we reveal in this manuscript.

Obviously, there is still a lot to do, future studies could focus on examining for example the direct interaction between O-GlcNAcylated Gal-1 and potential receptors on microglia. Additionally, investigating the downstream signaling pathways and cellular responses triggered by O-GlcNAcylation of Gal-1 could provide further insights into the mechanism underlying its regulation of microglia migration.

We appreciate the reviewer's feedback and will take this opportunity to further investigate the underlying mechanisms involved in the O-GlcNAcylation-mediated regulation of microglia migration in our future studies."

Point 2

3) The authors used 5 uM peptides and 30 minutes of incubation for microglial stimulation. The author should perform the dosing experiment, determine the best concentration and stimulation time, and show it in the result section.
Response: We had already established both the time (30 min) and the Aβ concentration (5µM) as optimal (Manterola, L. et al. Transl Psychiatry (2013) 3, e219; doi:10.1038/tp.2012.147; Wyssenbach, A et al. Aging Cell 2016 Dec;15(6):1140-1152.) to investigate early intracellular signaling in both neurons and astrocytes, and on this basis we have continued, in this case with microglia, and it works very well too.

- Authors should add these references in the manuscript.

Response: We have added them (see new references 23 and 25)

4) Which bands were used for quantification in Figure 1A?
Response: The bands between 72 and 55 Kd of lines 1 and 2.

-The authors should describe this in the manuscript.

Response: We have added it (see lines 4 and 5 of the legend of Figure 1)

Point 3

The authors showed that Abeta1-42 peptides increased O-GlcNacylation on Gal-1, which resulted in microglia migration. Is this phenomenon specific to the Abeta1-42 peptide? The authors should test if other peptides affect microglial migration as a control condition.

Response: We are not aware of it, but we believe that it is not a specific response to treatment with Aß, we think that Gal-1 O-GlcNAcylation is required to control migration, so probably other trophic factors that induce cell migration use this O-GlcNAcylation of the Gal-1 for its control.

-If the migration of microglia is not specific to the Abeta1-42 peptide, this manuscript would be misleading.

Response: I am very sorry that the reviewer thinks that if microglia migration is not Ab oligomers specific this manuscript would be misleading. I would like to clarify that the aim of this work was to investigate the role of Ab oligomers on the microglia biology, studying migration and the molecular mechanisms that control it. The question of whether or not it is a specific response of the Ab oligomers, the answer is we don't know. What we have revealed is that it is a response mediated by Ab oligomers. Going further, if the answer was specific to Ab oligomers it would be fine but if it were a universal answer for other factors, such as proinflammatory factors, chemokines, mitogenic peptides, etc, that would be fine too. The controls we have used are those propered by the experiments carried out in our work.

Point 4

5) No antibody information, anti-DsRed antibody in line 141, and specific antibodies in line 152. What is the dilution of antibody and incubation time, and what is the blocking buffer?
Response: In the new version we provide this information. The blocking buffer is a well-established standard buffer that can be purchased from Merck or Thermo Science that is used to block non-specific sites on blotting membranes. We think that it is not necessary to specify the components.
9) How to transfer the protein, and what membrane is used?
Response: The methodology that involves western blot is a very well established and standardized technique, so we do not believe that it is necessary to describe the technique in the manuscript.

-Authors responded that we think ..., we do not believe…, it is not the point. Authors should follow the instructions for authors. According to Materials and Methods in "instructions for authors": They should be described with sufficient detail to allow others to replicate and build on published results. New methods and protocols should be described in detail while well-established methods can be briefly described and appropriately cited. Give the name and version of any software used and make clear whether computer code used is available. Include any pre-registration codes.

Response: We have added in sections 2.7 and 2.8 of Materials and Methods the following sentence: …proteins were transferred to PVDF membranes (Life Science) then were blocked with 5% milk in TBST and finally
